# A hybrid sampler for Poisson-Kingman mixture models

**María Lomelí**
Gatsby Unit
University College London
mlomeli@gatsby.ucl.ac.uk

**Stefano Favaro**
Department of Economics and Statistics
University of Torino and Collegio Carlo Alberto
stefano.favaro@unito.it

**Yee Whye Teh**
Department of Statistics
University of Oxford
y.w.teh@stats.ox.ac.uk

## Abstract

This paper concerns the introduction of a new Markov Chain Monte Carlo scheme for posterior sampling in Bayesian nonparametric mixture models with priors that belong to the general Poisson-Kingman class. We present a novel compact way of representing the infinite dimensional component of the model such that while explicitly representing this infinite component it has less memory and storage requirements than previous MCMC schemes. We describe comparative simulation results demonstrating the efficacy of the proposed MCMC algorithm against existing marginal and conditional MCMC samplers.

## 1   Introduction

According to Ghahramani [9], models that have a nonparametric component give us more flexiblity that could lead to better predictive performance. This is because their capacity to learn does not saturate hence their predictions should continue to improve as we get more and more data. Furthermore, we are able to fully consider our uncertainty about predictions thanks to the Bayesian paradigm. However, a major impediment to the widespread use of Bayesian nonparametric models is the problem of inference. Over the years, many MCMC methods have been proposed to perform inference which usually rely on a tailored representation of the underlying process [5, 4, 18, 20, 28, 6]. This is an active research area since dealing with this infinite dimensional component forbids the direct use of standard simulation-based methods for posterior inference. These methods usually require a finite-dimensional representation. There are two main sampling approaches to facilitate simulation in the case of Bayesian nonparametric models: random truncation and marginalization. These two schemes are known in the literature as conditional and marginal samplers.

In conditional samplers, the infinite-dimensional prior is replaced by a finite-dimensional representation chosen according to a truncation level. In marginal samplers, the need to represent the infinite-dimensional component can be bypassed by marginalising it out. Marginal samplers have less storage requirements than conditional samplers but could potentially have worst mixing properties. However, not integrating out the infinite dimensional compnent leads to a more comprehensive representation of the random probability measure, useful to compute expectations of interest with respect to the posterior.

In this paper, we propose a novel MCMC sampler for Poisson-Kingman mixture models, a very large class of Bayesian nonparametric mixture models that encompass all previously explored ones in the literature. Our approach is based on a hybrid scheme that combines the main strengths of

both conditional and marginal samplers. In the flavour of probabilistic programming, we view our contribution as a step towards wider usage of flexible Bayesian nonparametric models, as it allows automated inference in probabilistic programs built out of a wide variety of Bayesian nonparametric building blocks.

## 2 Poisson-Kingman processes

Poisson-Kingman random probability measures (RPMs) have been introduced in Pitman [23] as a generalization of homogeneous Normalized Random Measures (NRMs) [25, 13]. Let $\mathbb{X}$ be a complete and separable metric space endowed with the Borel $\sigma$-field $\mathcal{B}(\mathbb{X})$, let $\mu \sim \mathrm{CRM}(\rho, H_0)$ be a homogeneous Completely Random Measure (CRM) with Lévy measure $\rho$ and base distribution $H_0$, see Kingman [15] for a good overview about CRMs and references therein. Then, the corresponding total mass of $\mu$ is $T = \mu(\mathbb{X})$ and let it be finite, positive almost surely, and absolutely continuous with respect to Lebesgue measure. For any $t \in \mathbb{R}^+$, let us consider the conditional distribution of $\mu/t$ given that the total mass $T \in dt$. This distribution is denoted by $\mathrm{PK}(\rho, \delta_t, H_0)$, it is the distribution of a RPM, where $\delta_t$ denotes the usual Dirac delta function. Poisson-Kingman RPMs form a class of RPMs whose distributions are obtained by mixing $\mathrm{PK}(\rho, \delta_t, H_0)$, over $t$, with respect to some distribution $\gamma$ on the positive real line. Specifically, a Poisson-Kingman RPM has following the hierarchical representation

$$
\begin{aligned}
T &\sim \gamma \\
P|T = t &\sim \mathrm{PK}(\rho, \delta_t, H_0).
\end{aligned}
\tag{1}
$$

The RPM $P$ is referred to as the Poisson-Kingman RPM with Lévy measure $\rho$, base distribution $H_0$ and mixing distribution $\gamma$. Throughout the paper we denote by $\mathrm{PK}(\rho, \gamma, H_0)$ the distribution of $P$ and, without loss of generality, we will assume that $\gamma(\mathrm{d}t) \propto h(t) f_\rho(t) \mathrm{d}t$ where $f_\rho$ is the density of the total mass $T$ under the CRM and $h$ is a non-negative function. Note that, when $\gamma(\mathrm{d}t) = f_\rho(t) \mathrm{d}t$ then the distribution $\mathrm{PK}(\rho, f_\rho, H_0)$ coincides with $\mathrm{NRM}(\rho, H_0)$. The resulting $P = \sum_{k \geqslant 1} p_k \delta_{\phi_k}$ is almost surely discrete and since $\mu$ is homogeneous, the atoms $(\phi_k)_{k \geqslant 1}$ of $P$ are independent of their masses $(p_k)_{k \geqslant 1}$ and form a sequence of independent random variables identically distributed according to $H_0$. Finally, the masses of $P$ have distribution governed by the Lévy measure $\rho$ and the distribution $\gamma$.

One nice property is that $P$ is almost surely discrete: if we obtain a sample $\{Y_i\}_{i=1}^n$ from it, there is a positive probability of $Y_i = Y_j$ for each pair of indexes $i \neq j$. Hence, it induces a random partition $\Pi$ on $\mathbb{N}$, where $i$ and $j$ are in the same block in $\Pi$ if and only if $Y_i = Y_j$. Kingman [16] showed that $\Pi$ is exchangeable, this property will be one of the main tools for the derivation of our hybrid sampler.

### 2.1 Size-biased sampling Poisson-Kingman processes

A second object induced by a Poisson-Kingman RPM is a size-biased permutation of its atoms. Specifically, order the blocks in $\Pi$ by increasing order of the least element in each block, and for each $k \in \mathbb{N}$ let $Z_k$ be the least element of the $k$th block. $Z_k$ is the index among $(Y_i)_{i \geqslant 1}$ of the first appearance of the $k$th unique value in the sequence. Let $\tilde{J}_k = \mu(\{Y_{Z_k}\})$ be the mass of the corresponding atom in $\mu$. Then $(\tilde{J}_k)_{k \geqslant 1}$ is a size-biased permutation of the masses of atoms in $\mu$, with larger masses tending to appear earlier in the sequence. It is easy to see that $\sum_{k \geqslant 1} \tilde{J}_k = T$, and that the sequence can be understood as a stick-breaking construction: starting with a stick of length $T_0 = T$; break off the first piece of length $\tilde{J}_1$; the surplus length of stick is $T_1 = T_0 - \tilde{J}_1$; then the second piece with length $\tilde{J}_2$ is broken off, etc.

Theorem 2.1 of Perman *et al.* [21] states that the sequence of surplus masses $(T_k)_{k \geqslant 0}$ forms a Markov chain and gives the corresponding initial distribution and transition kernels. The corresponding generative process for the sequence $(Y_i)_{i \geqslant 1}$ is as follows:

  i) Start with drawing the total mass from its distribution $\mathbb{P}_{\rho, h, H_0}(T \in dt) \propto h(t) f_\rho(t) dt$.
  ii) The first draw $Y_1$ from $P$ is a size-biased pick from the masses of $\mu$. The actual value of $Y_1$ is simply $Y_1^* \sim H_0$, while the mass of the corresponding atom in $\mu$ is $\tilde{J}_1$, with conditional

distribution

$$\mathbb{P}_{\rho,h,H_0}(\tilde{J}_1 \in ds_1 | T \in dt) = \frac{s_1}{t}\rho(ds_1)\frac{f_\rho(t-s_1)}{f_\rho(t)}, \quad \text{with surplus mass} \quad T_1 = T - \tilde{J}_1.$$

iii) For subsequent draws $i \geqslant 2$:
- Let $K$ be the current number of distinct values among $Y_1, \ldots, Y_{i-1}$, and $Y_1^*, \ldots, Y_K^*$ the unique values, i.e., atoms in $\mu$. The masses of these first $K$ atoms are denoted by $\tilde{J}_1, \ldots, \tilde{J}_K$ and the surplus mass is $T_K = T - \sum_{k=1}^K \tilde{J}_k$.
- For each $k \leqslant K$, with probability $\tilde{J}_k/T$, we set $Y_i = Y_k^*$.
- With probability $T_K/T$, $Y_i$ takes on the value of an atom in $\mu$ besides the first $K$ atoms. The actual value $Y_{K+1}^*$ is drawn from $H_0$, while its mass is drawn from

$$\mathbb{P}_{\rho,h,H_0}(\tilde{J}_{K+1} \in ds_{K+1} | T_K \in dt_K) = \frac{s_{K+1}}{t_K}\rho(ds_{K+1})\frac{f_\rho(t_K - s_{K+1})}{f_\rho(t_K)}, \quad T_{K+1} = T_K - \tilde{J}_{K+1}.$$

By multiplying the above infinitesimal probabilities, one obtains the joint distribution of the random elements $T$, $\Pi$, $(\tilde{J}_i)_{i \geqslant 1}$ and $(Y_i^*)_{i \geqslant 1}$

$$\mathbb{P}_{\rho,h,H_0}(\Pi_n = (c_k)_{k \in [K]}, Y_k^* \in dy_k^*, \tilde{J}_k \in ds_k \text{ for } k \in [K], T \in dt) \tag{2}$$

$$= t^{-n}f_\rho(t - \textstyle\sum_{k=1}^K s_k)h(t)dt \prod_{k=1}^K s_k^{|c_k|}\rho(ds_k)H_0(dy_k^*),$$

where $(c_k)_{k \in [K]}$ denotes a particular partition of $[n]$ with $K$ blocks, $c_1, \ldots, c_K$, ordered by increasing least element and $|c_k|$ is the cardinality of block $c_k$. The distribution (2) is invariant to the size-biased order. Such a joint distribution was first obtained in Pitman [23] , see also Pitman [24] for further details.

## 2.2    Relationship to the usual Stick-breaking construction

In the generative process above, we mentioned that it is reminiscent of the well known stick breaking construction from Ishwaran & James [12], where you break a stick of length one but it is not the same. However, we can effectively reparameterize the model, starting with Equation (2), due to two useful identities in distribution: $P_j \overset{d}{=} \frac{\tilde{J}_j}{T - \sum_{\ell < j} \tilde{J}_\ell}$ and $V_j \overset{d}{=} \frac{P_j}{1 - \sum_{\ell < j} P_\ell}$ for $j = 1, \ldots, K$. Indeed, using this reparameterization, we obtain the corresponding joint in terms of $K$ $(0,1)$-valued stick-breaking weights $\{V_j\}_{j=1}^K$ which correspond to a stick-breaking representation. Note that this joint distribution is for a general Lévy measure $\rho$, density $f_\rho$ and it is conditioned on the valued of the random variable $T$. We can recover the well known Stick breaking representations for the Dirichlet and Pitman-Yor processes, for a specific choice of $\rho$ and if we integrate out $T$, see the supplementary material for further details about the latter. However, in general, these stick-breaking random variables form a sequence of dependent random variables with a complicated distribution, except for the two previously mentioned processes, see Pitman [22] for details.

## 2.3    Poisson-Kingman mixture model

We are mainly interested in using Poisson-Kingman RPMs as a building block for an infinite mixture model. Indeed, we can use Equation (1) as the top level of the following hierarchical specification

$$\begin{aligned} T &\sim \gamma \\ P|T &\sim \mathrm{PK}(\rho_\sigma, \delta_T, H_0) \\ Y_i \mid P &\overset{\text{iid}}{\sim} P \\ X_i \mid Y_i &\overset{\text{ind}}{\sim} F(\cdot \mid Y_i) \end{aligned} \tag{3}$$

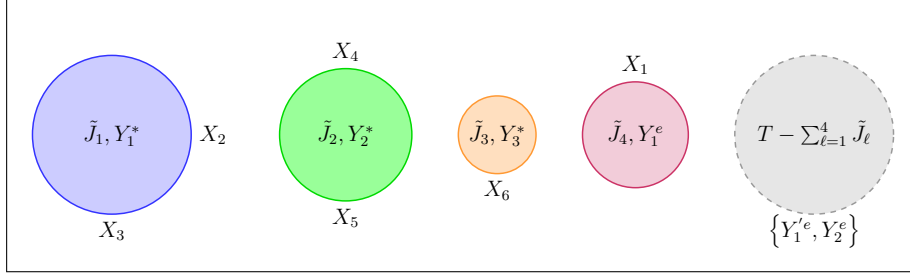

Figure 1: Varying table size Chinese restaurant representation for observations $\{X_i\}_{i=1}^9$

where $F(\cdot \mid Y)$ is the likelihood term for each mixture component, and our dataset consists of $n$ observations $(x_i)_{i\in[n]}$ of the corresponding variables $(X_i)_{i\in[n]}$. We will assume that $F(\cdot \mid Y)$ is smooth. After specifying the model we would like to carry out inference for clustering and/or density estimation tasks. We can do it exactly and more efficiently than with known MCMC samplers with our novel approach. In the next section, we present our main contribution and in the following one we show how it outperforms other samplers.

## 3 Hybrid Sampler

Equation's (2) joint distribution is written in terms of the first $K$ size-biased weights. In order to obtain a complete representation of the RPM, we need to size-bias sample from it a countably infinite number of times. Succesively, devise some way of representing this object exactly in a computer with finite memory and storage is needed.

We introduce the following novel strategy: starting from equation (2), we exploit the generative process of section 2.1 when reassigning observations to clusters. In addition to this, we reparameterize the model in terms of a surplus mass random variable $V = T - \sum_{k=1}^K \tilde{J}_k$ and end up with the following joint distribution

$$\mathbb{P}_{\rho,h,H_0}(\Pi_n = (c_k)_{k\in[K]}, Y_k^* \in dy_k^*, \tilde{J}_k \in ds_k \text{ for } k \in [K], T - \sum_{k=1}^K \tilde{J}_k \in dv, X_i \in dx_i \text{ for } i \in [n])$$

$$(4)$$

$$= (v + \sum_{k=1}^K s_k)^{-n} h\left(v + \sum_{k=1}^K s_k\right) f_\rho(v) \prod_{k=1}^K s_k^{|c_k|} \rho(ds_k) H_0(dy_k^*) \prod_{i\in c_k} F(dx_i|y_k^*).$$

For this reason, while having a complete representation of the infinite dimensional part of the model we only need to explicitly represent those size-biased weights associated to occupied clusters plus a surplus mass term which is associated to the rest of the empty clusters, as Figure 1 shows. The cluster reassignment step can be seen as a lazy sampling scheme: we explicitly represent and update the weights associated to occupied clusters and create a size-biased weight only when a new cluster appears. To make this possible we use the induced partition and we call Equation (4) the varying table size Chinese restaurant representation because the size-biased weights can be thought as the sizes of the tables in our restaurant. In the next subsection, we compute the complete conditionals of each random variable of interest to implement an overall Gibbs sampling MCMC scheme.

### 3.1 Complete conditionals

Starting from equation (4), we obtain the following complete conditionals for the Gibbs sampler

$$\mathbb{P}\left(V \in dv \mid \text{Rest}\right) \propto \left(v + \sum_{k=1}^K s_k\right)^{-n} f_\rho(v) h\left(v + \sum_{k=1}^K s_k\right) dv \tag{5}$$

$$\mathbb{P}\left(\tilde{J}_i \in ds_i \mid \text{Rest}\right) \propto \left(v + s_i + \sum_{k\neq i} s_k\right)^{-n} h\left(v + s_i + \sum_{k\neq i} s_k\right) s_i^{|c_i|} \rho(ds_i) \mathbb{I}_{(0,\text{Surpmass}_i)}(s_i) ds_i$$

where $\text{Surpmass}_i = V + \sum_{j=1}^{k} \tilde{J}_j - \sum_{j<i} \tilde{J}_j$.

$$\mathbb{P}(c_i = c \mid \mathbf{c}_{-i}, \text{Rest}) \propto \begin{cases} s_c F(dx_i \mid \{X_j\}_{j \in c} Y_c^*) & \text{if } i \text{ is assigned to existing cluster c} \\ \frac{v}{M} F(dx_i \mid Y_c^*) & \text{if } i \text{ is assigned to a new cluster c} \end{cases}$$

According to the rule above, the $i$th observation will be either reassigned to an existing cluster or to one of the $M$ new clusters in the ReUse algorithm as in Favaro & Teh [6]. If it is assigned to a new cluster, then we need to sample a new size-biased weight from the following

$$\mathbb{P}\left(\tilde{J}_{k+1} \in ds_{k+1} \mid \text{Rest}\right) \propto f_\rho(v - s_{k+1})\rho(s_{k+1})s_{k+1}\mathbb{I}_{(0,v)}(s_{k+1})ds_{k+1}. \tag{6}$$

Every time a new cluster is created we need to obtain its corresponding size-biased weight which could happen $1 \leqslant R \leqslant n$ times per iteration hence, it has a significant contribution to the overall computational cost. For this reason, an independent and identically distributed (i.i.d.) draw from its corresponding complete conditional (6) is highly desirable. In the next subsection we present a way to achieve this. Finally, for updating cluster parameters $\{Y_k^*\}_{k \in [K]}$, in the case where $H_0$ is non-conjugate to the likelihood, we use an extension of Favaro & Teh [6]'s ReUse algorithm, see Algorithm 3 in the supplementary material for details.

The complete conditionals in Equation (5) do not have a standard form but a generic MCMC method can be applied to sample from each within the Gibbs sampler. We use slice sampling from Neal [19] to update the size-biased weights and the surplus mass. However, there is a class of priors where the total mass's density is intractable so an additional step needs to be introduced to sample the surplus mass. In the next subsection we present two alternative ways to overcome this issue.

### 3.2 Example of classes of Poisson-Kingman priors

**a) $\sigma$-Stable Poisson-Kingman processes [23].** For any $\sigma \in (0, 1)$, let $f_\sigma(t) = \frac{1}{\pi}\sum_{j=0}^{\infty} \frac{(-1)^{j+1}}{j!}\sin(\pi\sigma j)\frac{\Gamma(\sigma j+1)}{t^{\sigma j+1}}$ be the density function of a positive $\sigma$-Stable random variable and $\rho(dx) = \rho_\sigma(dx) := \frac{\sigma}{\Gamma(1-\sigma)}x^{-\sigma-1}dx$. This class of RPMs is denoted by $\text{PK}(\rho_\sigma, h_T, H_0)$ where $h$ is a function that indexes each member of the class. For example, in the experimental section, we picked 3 choices of the $h$ function that index the following processes: Pitman-Yor, Normalized Stable and Normalized Generalized Gamma processes. This class includes all Gibbs type priors with parameter $\sigma \in (0, 1)$, so other choices of $h$ are possible, see Gnedin & Pitman [10] and De Blasi *et al.* [1] for a noteworthy account of this class of Bayesian nonparametric priors. In this case, the total mass's density is intractable and we propose two ways of dealing with this. Firstly, we used Kanter [14]'s integral representation for the $\sigma$-Stable density as in Lomeli *et al.* [17], introduce an auxiliary variable $Z$ and slice sample each variable

$$\mathbb{P}(V \in dv \mid \text{Rest}) \propto \left(v + \sum_{i=1}^{k} s_i\right)^{-n} v^{-\frac{\sigma}{1-\sigma}}\exp\left[-v^{\frac{-\sigma}{1-\sigma}}A(z)\right]h\left(v + \sum_{i=1}^{k} s_i\right)dv$$

$$\mathbb{P}(Z \in dz \mid \text{Rest}) \propto A(z)\exp\left[-v^{\left(-\frac{\sigma}{1-\sigma}\right)}A(z)\right]dz,$$

see Algorithm 1 in the supplementary material for details. Alternatively, we can completely bypass the evaluation of the total mass's density by updating the surplus mass with a Metropolis-Hastings step with an independent proposal from a Stable or from an Exponentially Tilted Stable($\lambda$). It is straight forward to obtain i.i.d draws from these proposals, see Devroye [3] and Hofert [11] for an improved rejection sampling method for the Exponentially tilted case. This leads to the following acceptance ratio

$$\frac{\mathbb{P}(V' \in dv' \mid \text{Rest}) f_\sigma(v)\exp(-\lambda v)}{\mathbb{P}(V \in dv \mid \text{Rest}) f_\sigma(v')\exp(-\lambda v')} = \frac{\left(v' + \sum_{i=1}^{k} s_i\right)^{-n} h\left(v' + \sum_{i=1}^{k} s_i\right)dv'\exp(-v)}{\left(v + \sum_{i=1}^{k} s_i\right)^{-n} h\left(v + \sum_{i=1}^{k} s_i\right)dv\exp(-v')},$$

see Algorithm 2 in the supplementary material for details. Finally, to sample a new size-biased weight

$$\mathbb{P}\left(\tilde{J}_{k+1} \in ds_{k+1} \mid \text{Rest}\right) \propto f_\sigma(v - s_{k+1})s_{k+1}^{-\sigma}\mathbb{I}_{(0,v)}(s_{k+1})ds_{k+1}.$$

Fortunately, we can get an i.i.d. draw from the above due to an identity in distribution given by Favaro *et al.* [8] for the usual stick breaking weights for any prior in this class such that $\sigma = \frac{u}{v}$ where $u < v$ are coprime integers. Then we just reparameterize it back to obtain the new size-biased weight, see Algorithm 4 in the supplementary material for details.

**b)** $-\log$**Beta-Poisson-Kingman processes [25, 27].** Let $f_\rho(t) = \frac{\Gamma(a+b)}{\Gamma(a)\Gamma(b)} \exp(-at)(1 - \exp(-t))^{b-1}$ be the density of a positive random variable $X \stackrel{d}{=} -\log Y$, where $Y \sim \text{Beta}(a, b)$ and $\rho(x) = \frac{\exp(-ax)(1-\exp(-bx))}{x(1-\exp(-x))}$. This class of RPMs generalises the Gamma process but has similar properties. Indeed, if we take $b = 1$ and the density function for $T$ is $\gamma(t) = f_\rho(t)$ we recover the Lévy measure and total mass's density function of a Gamma process. Finally, to sample a new size-biased weight

$$\mathbb{P}\left(\tilde{J}_{k+1} \in ds_{k+1} \mid \text{Rest}\right) \propto \frac{(1 - \exp(s_{k+1} - v))^{b-1}(1 - \exp(-bs_{k+1}))}{1 - \exp(-s_{k+1})} ds_{k+1} \mathbb{I}_{(0,v)}(s_{k+1}).$$

If $b > 1$, this complete conditional is a monotone decreasing unnormalised density with maximum at $b$. We can easily get an i.i.d. draw with a simple rejection sampler [2] where the rejection constant is $bv$ and the proposal is $U(0, v)$. There is no other known sampler for this process.

### 3.3 Relationship to marginal and conditional MCMC samplers

Starting from equation (2), another strategy would be to reparameterize the model in terms of the usual stick breaking weights. Next, we could choose a random truncation level and represent finitely many sticks as in Favaro & Walker [7]. Alternatively, we could integrate out the random probability measure and sample only the partition induced by it as in Lomeli *et al.* [17]. Conditional samplers have large memory requirements as often, the number of sticks needed can be very large. Furthermore, the conditional distributions of the stick lengths are quite involved so they tend to have slow running times. Marginal samplers have less storage requirements than conditional samplers but could potentially have worst mixing properties. For example, Lomeli *et al.* [17] had to introduce a number of auxiliary variables which worsen the mixing.

Our novel hybrid sampler exploits marginal and conditional samplers advantages. It has less memory requirements since it just represents the size-biased weights of occupied as opposed to conditional samplers which represent both empty and occupied clusters. Also, it does not integrate out the size-biased weights thus, we obtain a more comprehensive representation of the RPM.

## 4 Performance assesssment

We illustrate the performance of our hybrid sampler on a range of Bayesian nonparametric mixture models, obtained by different specifications of $\rho$ and $\gamma$, as in Equation (3). At the top level of this hierarchical specification, different Bayesian nonparametric priors were chosen from both classes presented in the examples section. We chose the base distribution $H_0$ and the likelihood term $F$ for the $k$th cluster to be

$$H_0(d\mu_k) = \mathcal{N}\left(d\mu_k \mid \mu_0, \sigma_0^2\right) \quad \text{and} \quad F(dx_1, \ldots, dx_{n_k} \mid \mu_k, \tau_1) = \prod_{i=1}^{n_k} \mathcal{N}\left(x_i \mid \mu_k, \sigma_1^2\right),$$

where $\{X_j\}_{j=1}^{n_k}$ are the $n_k$ observations assigned to the $k$th cluster at some iteration. $\mathcal{N}$ denotes a Normal distribution with mean $\mu_k$ and variance $\sigma_1^2$, a common parameter among all clusters. The mean's prior distribution is Normal, centered at $\mu_0$ and with variance $\sigma_0^2$. Although the base distribution is conjugate to the likelihood we treated it as non-conjugate case and sampled the parameters at each iteration rather than integrating them out.

We used the dataset from Roeder [26] to test the algorithmic performance in terms of running time and effective sample size (ESS), as Table 1 shows. The dataset consists of measurements of velocities in km/sec of $n = 82$ galaxies from a survey of the Corona Borealis region. For the $\sigma$-Stable Poisson-Kingman class, we compared it against our implementation of Favaro & Walker [7]'s conditional sampler and against the marginal sampler of Lomeli *et al.* [17]. We chose to compare our hybrid sampler against these existing approaches which follow the same general purpose paradigm.

| Algorithm | $\sigma$ | Running time | ESS($\pm$std) |
|---|---|---|---|
| Pitman-Yor process ($\theta = 10$) | | | |
| Hybrid | 0.3 | 7135.1(28.316) | **2635.488(187.335)** |
| Hybrid-MH ($\lambda = 0$) | 0.3 | 5469.4(186.066) | 2015.625(152.030) |
| Conditional | 0.3 | NA | NA |
| Marginal | 0.3 | **4685.7(84.104)** | 2382.799(169.359) |
| Hybrid | 0.5 | **3246.9(24.894)** | **3595.508(174.075)** |
| Hybrid-MH ($\lambda = 50$) | 0.5 | 4902.3(6.936) | 3579.686(135.726) |
| Conditional | 0.5 | 10141.6(237.735) | 905.444(41.475) |
| Marginal | 0.5 | 4757.2(37.077) | 2944.065(195.011) |
| Normalized Stable process | | | |
| Hybrid | 0.3 | **5054.7(70.675)** | **5324.146(167.843)** |
| Hybrid-MH ($\lambda = 0$) | 0.3 | 7866.4(803.228) | 5074.909(100.300) |
| Conditional | 0.3 | NA | NA |
| Marginal | 0.3 | 7658.3(193.773) | 2630.264(429.877) |
| Hybrid | 0.5 | 5382.9(57.561) | **4877.378(469.794)** |
| Hybrid-MH ($\lambda = 50$) | 0.5 | **4537.2(37.292)** | 4454.999(348.356) |
| Conditional | 0.5 | 10033.1(22.647) | 912.382(167.089) |
| Marginal | 0.5 | 8203.1(106.798) | 3139.412(351.788) |
| Normalized Generalized Gamma process ($\tau = 1$) | | | |
| Hybrid | 0.3 | **4157.8(92.863)** | **5104.713(200.949)** |
| Hybrid-MH ($\lambda = 0$) | 0.3 | 4745.5(187.506) | 4848.560(312.820) |
| Conditional | 0.3 | NA | NA |
| Marginal | 0.3 | 7685.8(208.98) | 3587.733(569.984) |
| Hybrid | 0.5 | 6299.2(102.853) | **4646.987(370.955)** |
| Hybrid-MH ($\lambda = 50$) | 0.5 | **4686.4(35.661)** | 4343.555(173.113) |
| Conditional | 0.5 | 10046.9(206.538) | 1000.214(70.148) |
| Marginal | 0.5 | 8055.6(93.164) | 4443.905(367.297) |
| -logBeta ($a = 1, b = 2$) | | | |
| Hybrid | - | **2520.6(121.044)** | **3068.174(540.111)** |
| Conditional | - | NA | NA |
| Marginal | - | NA | NA |

Table 1: Running times in seconds and ESS averaged over 10 chains, 30,000 iterations, 10,000 burn in.

Table 1 shows that different choices of $\sigma$ result in differences in the algorithm's running times and ESS. The reason for this is that in the $\sigma = 0.5$ case there are readily available random number generators which do not increase the computational cost. In contrast, in the $\sigma = 0.3$ case, a rejection sampler method is needed every time a new size-biased weight is sampled which increases the computational cost, see Favaro *et al.* [8] for details. Even so, in most cases, we outperform both marginal and conditional MCMC schemes in terms of running times and in all cases, in terms of ESS. In the Hybrid-MH case, even thought the ESS and running times are competitive, we found that the acceptance rate is not optimal, we are currently exploring other choices of proposals. Finally, in Example b), our approach is the only one available and it has good running times and ESS. This qualitative comparison confirms our previous statements about our novel approach.

## 5  Discussion

Our main contribution is our Hybrid MCMC sampler as a general purpose tool for inference with a very large class of infinite mixture models. We argue in favour of an approach in which a generic algorithm can be applied to a very large class of models, so that the modeller has a lot of flexibility in choosing specific models suitable for his/her problem of interest. Our method is a hybrid approach since it combines the perks of the conditional and marginal schemes. Indeed, our experiments confirm that our hybrid sampler is more efficient since it outperforms both marginal and conditional samplers in running times in most cases and in ESS in all cases.

We introduced a new compact way of representing the infinite dimensional component such that it is feasible to perform inference and how to deal with the corresponding intractabilities. However, there are still various challenges that remain when dealing with these type of models. For instance, there are some values for $\sigma$ which we are unable to perform inference with our novel sampler. Secondly, when a Metropolis-Hastings step is used, there could be other ways to improve the mixing in terms of better proposals. Finally, all BNP MCMC methods can be affected by the dimensionality and size of the dataset when dealing with an infinite mixture model. Indeed, all methods rely on the same way of dealing with the likelihood term. When adding a new cluster, all methods sample its

corresponding parameter from the prior distribution. In a high dimensional scenario, it could be very difficult to sample parameter values close to the existing data points. We consider these points to be an interesting avenue of future research.

### Acknowledgments

We thank Konstantina Palla for her insightful comments. María Lomelí is funded by the Gatsby Charitable Foundation, Stefano Favaro is supported by the European Research Council through StG N-BNP 306406 and Yee Whye Teh is supported by the European Research Council under the European Unions Seventh Framework Programme (FP7/2007-2013) ERC grant agreement no. 617071.

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
