[Supplementary Material]

# Supplementary material: a hybrid sampler for Poisson-Kingman mixture models

**María Lomeli**
Gatsby Unit
University College London
mlomeli@gatsby.ucl.ac.uk

**Stefano Favaro**
Department of Economics and Statistics
University of Torino and Collegio Carlo Alberto
stefano.favaro@unito.it

**Yee Whye Teh**
Department of Statistics
University of Oxford
y.w.teh@stats.ox.ac.uk

## 1 Pseudocode

---

**Algorithm 1** HybridSampler$\sigma$-PK$\Big(K, V, \mathbf{c}, \{X_i\}_{i\in[n]}, \{Y_c^*\}_{c\in\Pi_n}, \{\tilde{J}_c\}_{c\in\Pi_n}, H_0, M\Big)$

---

**for** $t = 2 \to iter$ **do**
    Update $v^{(t)}$: Slice sample $\tilde{\mathbb{P}}\left(V \in \mathrm{d}v \mid \text{rest}\right)$
    Update $s_i^{(t)}$ for $i = 1, \ldots, k$ : Slice sample $\tilde{\mathbb{P}}\left(\tilde{J}_i \in \mathrm{d}s_i \mid \text{rest}\right)$
    Update $\pi^{(t)}, \left\{y_c^{*(t)}\right\}_{c\in\pi}, \left\{s_c^{(t)}\right\}_{c\in\pi}$ : **AddTable&ReUse**$\Big(V, \Pi_n, M, \{X_i\}_{i\in[n]}, \{Y_c^*\}_{c\in\Pi_n}, \{\tilde{J}_c\}_{c\in\Pi_n}, H_0 \mid \text{rest}\Big)$
**end for**

---

---

**Algorithm 2** HybridSampler-MH-$\sigma$PK$\Big(K, V, \mathbf{c}, \{X_i\}_{i\in[n]}, \{Y_c^*\}_{c\in\Pi_n}, \{\tilde{J}_c\}_{c\in\Pi_n}, H_0, M\Big)$

---

**for** $t = 2 \to iter$ **do**
    Update $s_i^{(t)}$ for $i = 1, \ldots, k$ : Slice sample $\tilde{\mathbb{P}}\left(\tilde{J}_i \in \mathrm{d}s_i \mid \text{rest}\right)$
    Update $v^{(t)}$: MH step for $\tilde{\mathbb{P}}\left(V \in \mathrm{d}v \mid \text{rest}\right)$ with independent proposal `Stablernd`$(\sigma)$ or `ExpTiltStablernd`$(\lambda, \sigma)$.
    Update $\pi^{(t)}, \left\{y_c^{*(t)}\right\}_{c\in\pi}, \left\{s_c^{(t)}\right\}_{c\in\pi}$ : **AddTable&ReUse**$\Big(V, \Pi_n, M, \{X_i\}_{i\in[n]}, \{Y_c^*\}_{c\in\Pi_n}, \{\tilde{J}_c\}_{c\in\Pi_n}, H_0 \mid \text{rest}\Big)$
**end for**

---

---

**Algorithm 3** AddTable&ReUse$\left(V, \Pi_n, M, \{X_i\}_{i\in[n]}, \{Y_c^*\}_{c\in\Pi_n}, \{\tilde{J}_c\}_{c\in\Pi_n}, H_0 \mid \text{rest}\right)$

---

Let $c \in \Pi_n$ be such that $i \in c$
$c \leftarrow c\backslash\{i\}$
**if** $c = \varnothing$ **then**
$\quad k \sim \text{UniformDiscrete}(\frac{1}{M})$
$\quad Y_k^e \leftarrow Y_c^*$
$\quad \Pi_n \leftarrow \Pi_n\backslash\{c\}$
$\quad V \leftarrow V + \tilde{J}_c$ $\qquad\qquad\qquad\qquad\qquad$ ▷ Add back the discarded table size to the surplus
**end if**

Set $c'$according to$\mathbb{P}(c_i = c \mid \mathbf{c}_{-i}, \text{Rest}) \propto \begin{cases} \tilde{J}_c F(x_i \mid \{X_i\}_{i\in c} \, Y_c^*) & \text{if existing} \\ \frac{V}{M} F(x_i \mid Y_c^*) & \text{if new} \end{cases}$

**if** $c' \in [M]$ **then**
$\quad \tilde{J}_{\text{new}} \leftarrow$ **ExactSampleNewTableSize**$(V, \sigma \mid \text{rest})$
$\quad V \leftarrow V - \tilde{J}_{\text{new}}$ $\qquad\qquad\qquad\qquad\qquad$ ▷ Remove it from the old surplus
$\quad \Pi_n \leftarrow \Pi_n \cup \{\{i\}\}$
$\quad Y_{\{i\}}^* \leftarrow Y_{c'}^e$
$\quad Y_{c'}^e \sim H_0$
**else**
$\quad c' \leftarrow c' \cup \{i\}$
**end if**
Draw $\{Y_j^e\}_{j=1}^M \overset{\text{i.i.d.}}{\sim} H_0$

---

**Algorithm 4** ExactSampleNewTableSize$(V, \sigma \mid \text{rest})$

---

**if** $\sigma = 0.5$ **then**
$\quad G \sim \text{Gamma}\left(\frac{3}{4}, 1\right)$
$\quad IG \sim \text{Inverse Gamma}\left(\frac{1}{4}, \frac{1}{4^3}V^{-2}\right)$
$\quad V_{stick} = \frac{\sqrt{G}}{\sqrt{G}+\sqrt{IG}}$
$\quad \tilde{J}_{new} = V_{stick}V$
**else**
$\quad$ **if** $\sigma < 0.5 \ \&\& \ \sigma = \frac{u_\sigma}{v_\sigma}, u_\sigma, v_\sigma \in \mathbb{N}$ **then**
$\quad\quad \lambda = u_\sigma^2/v_\sigma^{\frac{v_\sigma}{u_\sigma}}$
$\quad\quad IG \sim \text{Inverse Gamma}\left(1 - \frac{u_\sigma}{v_\sigma}, \lambda\right)$
$\quad\quad \frac{1}{G} \sim \mathcal{E}_{\mathcal{T}}\left(\lambda, L_{\frac{u_\sigma}{v_\sigma}}^{-1/u}\right)$ $\qquad$ ▷ Samples an exponentially tilted random variable. See Favaro
$\quad$ *et al.* [1] for details.
$\quad\quad V_{stick} = \frac{G}{G+IG}$
$\quad\quad \tilde{J}_{new} = V_{stick}V$
$\quad$ **end if**
**end if**

---

## 2 Relationship to the Pitman-Yor's Stick-breaking construction

Figure 1: Generative process of Section 2.1

Figure 2: Pitman-Yor's stick breaking construction

$$T \sim \gamma_{\mathrm{PY}}$$
$$\tilde{J}_1 \mid T \sim \mathrm{SBS}(T)$$
$$\tilde{J}_2 \mid T, \tilde{J}_1 \sim \mathrm{SBS}\left(T - \tilde{J}_1\right)$$
$$\vdots$$
$$\tilde{J}_\ell \mid T, \tilde{J}_1, \ldots, \tilde{J}_{\ell-1} \sim \mathrm{SBS}\left(T - \sum_{i<\ell} \tilde{J}_i\right)$$
$$\vdots$$
$$P_\ell \stackrel{d}{=} \frac{\tilde{J}_\ell}{T - \sum_{j<\ell} \tilde{J}_j}$$

$$V_1 \sim \mathrm{Beta}(v_1 \mid 1 - \sigma, \theta + \sigma)$$
$$V_2 \sim \mathrm{Beta}(v_2 \mid 1 - \sigma, \theta + 2\sigma)$$
$$\vdots$$
$$V_\ell \sim \mathrm{Beta}(v_\ell \mid 1 - \sigma, \theta + \ell\sigma)$$
$$\vdots$$

the corresponding weights are:

$$P_\ell \stackrel{d}{=} V_\ell \prod_{j<\ell}(1 - V_j).$$

The Pitman Yor's stick breaking construction from Ishwaran & James [2] can be recovered from the size-biased sampling (SBS) generative process of Section 2.1 after integrating out the total mass $T$, the change of variables given in Section 2.2 and if we choose a specific distribution for the total mass

$$\gamma_{\mathrm{PY}}(t) = \frac{\Gamma(\theta + 1)}{\Gamma(\frac{\theta}{\sigma} + 1)} t^{-\theta} f_\sigma(t) \mathbb{I}_{(0,\infty)}(t), \qquad \theta > -\sigma$$

which corresponds to the Pitman-Yor prior and $f_\sigma$ is the density of a $\sigma$-Stable random variable.