[Reviews · NeurIPS 2015]

Submitted by Assigned_Reviewer_1

Summary:

The authors present a new MCMC method for a general class of bayesian nonparametric models, namely Poisson-Kingman random probability measures.

They describe the class of models, the generative procedure and the insight that yields an mcmc routine for inference, as well as existing methods for the same/similar class of models, namely marginal and conditional samplers. They compare their sampler to standard marginal and conditional samplers, comparing effective sample size and running time.

- Quality

This paper develops an inference method for a very general class of models.

It seems technically sound and provides an adequate summary of existing methods and how they relate to the method they present.

- Clarity

This paper isn't that clearly written, which may be due to the page limit.

For instance, it is very difficult to decipher the notation in the generative procedure of section 2.1, which is arguably the most important section for appreciating the authors' contribution.

- Originality

It is my understanding that a sampler for this general class of models is novel.

- Significance

If the idea were presented more fluidly, this paper could serve as a nice review and reference for this type of sampler.

I think the idea has the potential to be impactful, but may fall short due to presentation.

Comments:

- A figure or diagram of your generative process (the stick breaking construction) and how it is distinct from the standard DP stick breaking construction of Ishwaran and James would make section 2 (as well as your methodological contribution in general) easier to understand.

Summary: This paper seems technically sound and potentially impactful, but needs work on presentation of its ideas.

Submitted by Assigned_Reviewer_2

Overall I liked the paper although at times it is difficult to follow. The authors make a good effort at going from the process interpretation to the generative model interpretation and also repeatedly pointed out the main goal to achieve by their proposed method. If they can match it with extensive experiments I think this will be a great paper.

The authors have answered my questions on experiments in their rebuttal and I hope that these details will be added in the final version.
Summary: The paper presents a Hybrid MCMC sampler for Poisson-Kingman mixture model. It combines the usefulness of conditional and marginal sampler. The authors claim that this reduces the memory requirement while increasing mixing.

My main criticism is with respect to the experiments done in this paper. They have shown result on uni-dimensional dataset without any experiments to show that their method is not effected with increasing dimension or size of datasets. Nor is their any experiment to show how significant of a reduction in memory requirement they get which has been mentioned as one of the main reasons for which this method should be used. This is why I did not give this paper a 7 score. I would request the authors to comment on this.

Submitted by Assigned_Reviewer_3

The paper proposes a new sampler for mixture models based on homogeneous completely random measures. The sampler is a hybrid of two existing classes of samplers and retains the favorable properties of each: the faster mixing times of the so-called conditional samplers, and the ability to exactly represent an infinite object with finite memory of the so-called marginal samplers. The paper uses results from Pitman's work on Poisson-Kingman partitions and size-biased sampling in order to derive a Gibbs sampling scheme for Poisson-Kingman mixture models. The authors show that the proposed sampler performs favorably, in terms of run time and of effective sample size, as compared to the existing conditional and marginal samplers.

Mixture models based on normalized CRMs have received an increasing amount of attention in the Bayesian non-parametrics literature in recent years, and the sampler proposed in this paper is a valuable tool for the community. The key technical insight is to represent the "surplus mass" as a variable in the sampler, and to obtain its complete conditional distribution by using Pitman's results on the joint distribution of the weights and the total mass of a CRM. The exposition is clear and the work is significant for the Bayesian non-parametrics community.
Summary: The paper proposes an efficient sampler for a class of models that has received increasing attention in recent years. It is well-executed and clearly written, and provides an important tool for the Bayesian non-parametrics community.

Author Feedback
Author rebuttal: We thank the reviewers for their careful reading of the manuscript and insightful comments.

R1: We agree with the remark, there are similarities between the two stick breaking constructions. However, the generative process of section 2.1 and Pitman (2003) is more general, it is valid for any member of the class of Poisson Kingman processes (PKP). Most members of this class do not have a stick breaking construction in terms of independent random variables, see lines 135-136 and Pitman (1996) for further details. However, for the Pitman Yor process, the stick breaking construction does reduce to the one in Ishwaran and James (2001) after integrating out the random variable T and a change of variables given in line 128. We thank reviewer 1 for the figure suggestion, we will add it to the final version if the paper gets accepted.

R2: We agree that our method, in fact, all Bayesian nonparametric Markov chain Monte Carlo (BNP MCMC) methods, can be affected by increasing the size of the dataset n. In lines 212-213, we state the worst case scenario: when updating the cluster assignment for each observation, we would have to store at most n auxiliary variables i.e. each observation is assigned to a cluster by its own. This worst case scenario can happen with all BNP MCMC methods, it should not be seen as a unique drawback of our method. Even though the worst case can happen, it is usually the case that the number of auxiliary variables (in both marginal and hybrid samplers) is K < n due to the almost sure discreteness of the underlying random probability measure. Furthermore, the asymptotic growth rate for all \sigma-Stable PKP is K_n\sim n^{\sigma}T^{-\sigma} as n\rightarrow \infty.

We retain there is no needed to verify the less storage requirements experimentally. In line 314, we mention the main difference with the slice conditional sampler of Favaro and Walker (2013): this method could have large storage requirements when the truncation level is very low, see Walker (2007) and Kalli, Griffiths and Walker (2011)for details. This means a number of occupied clusters plus a number of empty clusters (a random nonnegative quantity E) needs to be stored. Hence, the memory requirement per iteration is greater than the worst case for the hybrid sampler, i.e. n+E > n.

We also agree that the method in fact, all BNP MCMC methods, can be affected by the dimensionality of the dataset, we consider this to be an orthogonal problem to the one we tackle. For example, when dealing with an infinite mixture model with a multidimensional likelihood component. This is because all methods rely on the same way of dealing with the likelihood term. For instance, when adding a new cluster, all methods sample its corresponding parameter from the prior distribution. In a high dimensional scenario, it could be very difficult to sample parameter values close to the existing data points, all methods suffer from the curse of dimensionality. One solution is to preprocess the data to capture the most relevant dimensions with some dimensionality reduction technique and run the samplers on the projected dataset. There could be more sophisticated ways of solving this issue, we recognise this is an interesting avenue of research, it is beyond the scope of the present paper.

Finally, we agree that exploring the performance of the sampler on other datasets is a topic of interest, this is a route we are currently pursuing. Some preliminary experiments on an 4000-dimensional dataset used in Lomeli et al (2014). We preprocessed the data with PCA and ran the MCMC on the projected data, d = 8, 30,000 iterations, 10,000 burn in:
Running times | ESS

PY

Hybrid | 4.410e+03 (400.2) | 56.736 (40.908)
Marginal | 6.417e+04 (576.5) | 86.120 (22.875)

NGG

Hybrid | 6.862e+03 (327.70) | 1516.087 (3136.666)
Marginal | 5.341e+04 (1911.8) | 148.062 (44.294)

NS

Hybrid | 6.441e+03 (428.4) | 112.331 (124.664)
Marginal | 6.359e+04 (562.7) | 92.937 (88.145)

We can see how for both MCMC samplers the ESS is significantly affected by the dimensionality of the problem. However, our hybrid sampler outperforms the marginal in terms of running times by an order of magnitude.

R3: We thank reviewer 3 for pointing out the typo in equation (2). However, the reviewer's first comment is incorrect. The h function only affects the distribution of the total mass so we only need it when we draw the total mass initially. You can see this from the generative process of section 2.1 and Pitman (2003).

We thank Reviewers 4, 5 and 6 for their very positive assessment of the manuscript and Reviewer 1, 2 and 3 for their observant suggestions that will be incorporated in the camera-ready version, improving the quality of the manuscript.